# An extremely fast neural mechanism to detect emotional visual stimuli: A two-experiment study

**Luis Carretié**©\*, **Uxía Fernández-Folgueiras**©, **Dominique Kessel**©, **Guzmán Alba**©, **Estrella Veiga-Zarza**©, **Manuel Tapia**, **Fátima Álvarez**©

Facultad de Psicología, Universidad Autónoma de Madrid, Madrid, Spain

\* carretie@uam.es

**Data Availability Statement:** Data described in the paper are available at the Open Science Framework (https://osf.io/9bc2y).

## Abstract

Defining the brain mechanisms underlying initial emotional evaluation is a key but unexplored clue to understanding affective processing. Event-related potentials (ERPs), especially suited for investigating this issue, were recorded in two experiments (n = 36 and n = 35). We presented emotionally negative (spiders) and neutral (wheels) silhouettes homogenized regarding their visual parameters. In Experiment 1, stimuli appeared at fixation or in the periphery (200 trials per condition and location), the former eliciting a N40 (39 milliseconds) and a P80 (or C1: 80 milliseconds) component, and the latter only a P80. In Experiment 2, stimuli were presented only at fixation (500 trials per condition). Again, an N40 (45 milliseconds) was observed, followed by a P100 (or P1: 105 milliseconds). Analyses revealed significantly greater N40-C1P1 peak-to-peak amplitudes for spiders in both experiments, and ANCOVAs showed that these effects were not explained by C1P1 alone, but that processes underlying N40 significantly contributed. Source analyses pointed to V1 as an N40 focus (more clearly in Experiment 2). Sources for C1P1 included V1 (P80) and V2/LOC (P80 and P100). These results and their timing point to low-order structures (such as visual thalamic nuclei or superior colliculi) or the visual cortex itself, as candidates for initial evaluation structures.

## Introduction

Despite the growing interest and knowledge on the neural mechanisms sustaining emotional processing, several basic, key issues remain far from being understood. One of them, especially relevant in evolutionary terms, is how the brain deals so rapidly with emotional stimulation, organizing behavioral reactions that, in some circumstances, occur within four or five-tenths of a second (e.g., [1]). An obvious and necessary previous neural process is detecting emotional stimuli or, in other words, initially evaluating the incoming sensory input and marking it, if pertinent, as dangerous, appetitive, or, in general, affectively loaded. These initial evaluation structures (IESs) are still undefined, despite several hypotheses have been proposed. An intimately related and unsolved issue is the latency at which these structures can elicit electrophysiological traces, by themselves or through their cortical projections, of their

**Funding:** This work was supported by Ministerio de Ciencia e Innovación [MICINN; grant number PID2021-124420NB-I00).

**Competing interests:** The authors have declared that no competing interests exist.

evaluative activity. Crucially, this latency may help to reinforce some of the proposals on IESs over others.

For example, the detection of subliminal facial expressions by the amygdala, often conceptualized as a core IES (e.g., see reviews or meta-analyses in [2–5]), lasts more than 250 milliseconds (ms) to be reflected in the visual cortex [6]. However, the visual cortex shows greater activity to emotional stimuli than to neutral ones before 100 ms [7]. Thus, it is improbable that the amygdala mediates this early visual cortical activity. Moreover, while emotional faces elicit an increased response in the amygdala as compared to neutral faces as soon as 74 ms, amygdalar discrimination of emotional *non-facial* stimuli occurs beyond 150 ms from stimulus onset [8]. Again, evidence exists of visual cortex discrimination of emotional non-facial stimuli before 100 ms [7]. In other words, current data point to candidates other than the amygdala to be IESs. An alternative hypothesis is that initial -albeit rudimentary- evaluation may reside in faster ($\approx$30- ms) first-order structures (i.e., receiving direct visual inputs from the retina). This low-order IES hypothesis [9] points to the visual thalamus, mainly the lateral geniculate nucleus (LGN)-thalamic reticular nucleus (TRN) tandem, which are recently being revealed as active processors of the visual input rather than passive relays, as traditionally assumed (see reviews in [10, 11]), with the contribution of other first- and second-order thalamic and non-thalamic nuclei such as the superior colliculus (e.g., [12]). Finally, a third hypothesis is that the visual cortex itself is the initial evaluator, its activity being not relevantly mediated by any previous evaluation process [13].

Thus, the latency of the first biased response to emotional stimuli in the visual cortex indirectly informs on the nature of the IES involved. Particularly, shorter latencies would support non-amygdalar hypotheses. The best non-invasive methodologies to study response latencies of human neural processes are magnetoelectric (EEG and MEG), whose temporal resolution is much higher than that of the rest of non-invasive neuroimaging techniques. In the visual domain, the earliest trace in event-related potentials (ERPs) -one of the neural signals the EEG provides- signaling cortical processing is usually attributed to the C1 component. This component initiates at $\approx$60 ms from stimulus onset and peaks at $\approx$80, and is mainly originated in the striate cortex or V1 [14, 15], although the contribution of V2 and V3 has also been raised [14, 16]. Importantly, C1 peak shows enhanced amplitudes in response to emotional facial expressions as compared to neutral [17–20]. Non-facial, consciously perceived emotional stimuli of pictorial nature have not been explored in this respect, but emotional effects have been observed in response to affective words between 50 and 100 ms [20].

Although less frequently, visual ERP components before C1 have also been reported in the research domain [21–27]. It is important to note, however, that visual ERP components as early as 30 ms are well established in clinical practice in response to visual flashes [28]. This pre-C1 activity, which lacks a consensual nomenclature, will be referred to as N40 hereafter. The scarce data available on the origin of N40 point to V1 as one of its sources. Indeed, the onset of V1 activity once the geniculo-cortical inputs arrive may occur as early as 18–20 ms in the macaque monkey [29, 30] with an average latency reported at 26 ms [30]. An extrapolation to humans following the rough 3/5 ratio characterizing macaque vs human latencies (e.g., [31]) yields an approximate latency of 40 ms in our species. The contribution of V1 to the generation of N40 has also been reported in humans [32] along with thalamic sources [33]. Interestingly, N40 has been revealed to be modulated by attention, so it appears to be sensitive to cognitive factors [25, 33]. Modulation of such an early ERP activity by emotional stimuli has not been explored yet, however. Thus, the scarce studies exploring these extremely early electrophysiological traces of visual processing employ non-emotional stimuli. On the other hand, ERP studies presenting emotional stimuli have not been designed to explore N40, which presents a relatively low amplitude and hence, low signal-to-noise ratio (SNR).

We carried out two experiments to explore whether ERP activity originating in the visual cortex during the first 100 ms is enhanced by non-facial, supraliminal, emotional visual stimuli in order to advance in the characterization of IESs. To this aim, we introduced several methodological implementations that help to enhance the SNR in early visual ERP components. First, the number of trials was larger than usual in ERP research on emotional stimulation. Second, stimuli presented Gestalt characteristics such as closed contours or compact shape (they consisted of silhouettes) since they are optimal to increase the response of contour-sensitive neurons present in V1 and V2 (e.g. [34]). Third, in Experiment 1, stimuli were presented at several spatial locations given that cognitive (attentional) and emotional effects on early visual ERP components (such as C1) may be modulated (and even neutralized) depending on their position in the visual field (attentional effects: [35]; emotional: [7]). Experiment 2 employed only stimuli at fixation given the results of Experiment 1, and this allowed for a further increase in the number of trials. This second experiment also included several design modifications to control for potential alternative explanations of the effects observed in Experiment 1.

## Materials and methods

### Data and supplemental material availability

The data associated with both experiments are available at https://osf.io/9bc2y. Supplemental material mentioned hereafter is also available at that link.

### Participants

Forty-four individuals participated in Experiment 1, although data from only 36 of them could eventually be analyzed, as explained later. These 36 participants (age range of 18 to 24 years, mean = 19.46, SD = 1.14, 29 women) were students of Psychology, provided their written informed consent, and received academic compensation for their participation. In the case of Experiment 2, the initial sample consisted of thirty-nine individuals, none of whom participated in Experiment 1. The data from only 35 of them (age range of 17 to 24 years, mean = 19.68, SD = 2.11, 30 women) could eventually be analyzed, as explained later. They also provided their written informed consent (that of the only participant under the legal age of majority in Spain -18 years- was also signed by one of the parents).

In both cases, sample sizes allow reaching a statistical power of at least 0.8 for two dependent means comparisons -spiders vs. wheels, in this case- foreseeing medium effect sizes, usual in studies on early ERPs (computations were carried out employing G*Power© developed by [36]). Both experiments were designed in accordance with the Declaration of Helsinki and had been previously approved by the Universidad Autónoma de Madrid's Ethics Committee. The whole sample of participants attended the laboratory between November 10 and 30, 2020 (Experiment 1) and between September 18 and November 20, 2023 (Experiment 2).

### Stimuli (common to Experiments 1 and 2)

Participants were placed in an electrically shielded, sound-attenuated room. They were asked to place their chin on a chinrest maintained at a fixed distance (40 cm) from the screen (VIEWpixx®, 120 Hz) throughout the experiment. The presentation software was Psychtoolbox 3, and the communication between the stimulation PC and the Biosemi© EEG recording system was via optic fiber. The inevitable lag between the marks signaling stimuli onsets (or 'triggers') in EEG recordings and its actual onset on the screen was measured employing a photoelectric sensor as described in https://www.youtube.com/watch?v=0BPwcciq8u8 and corrected during pre-processing.

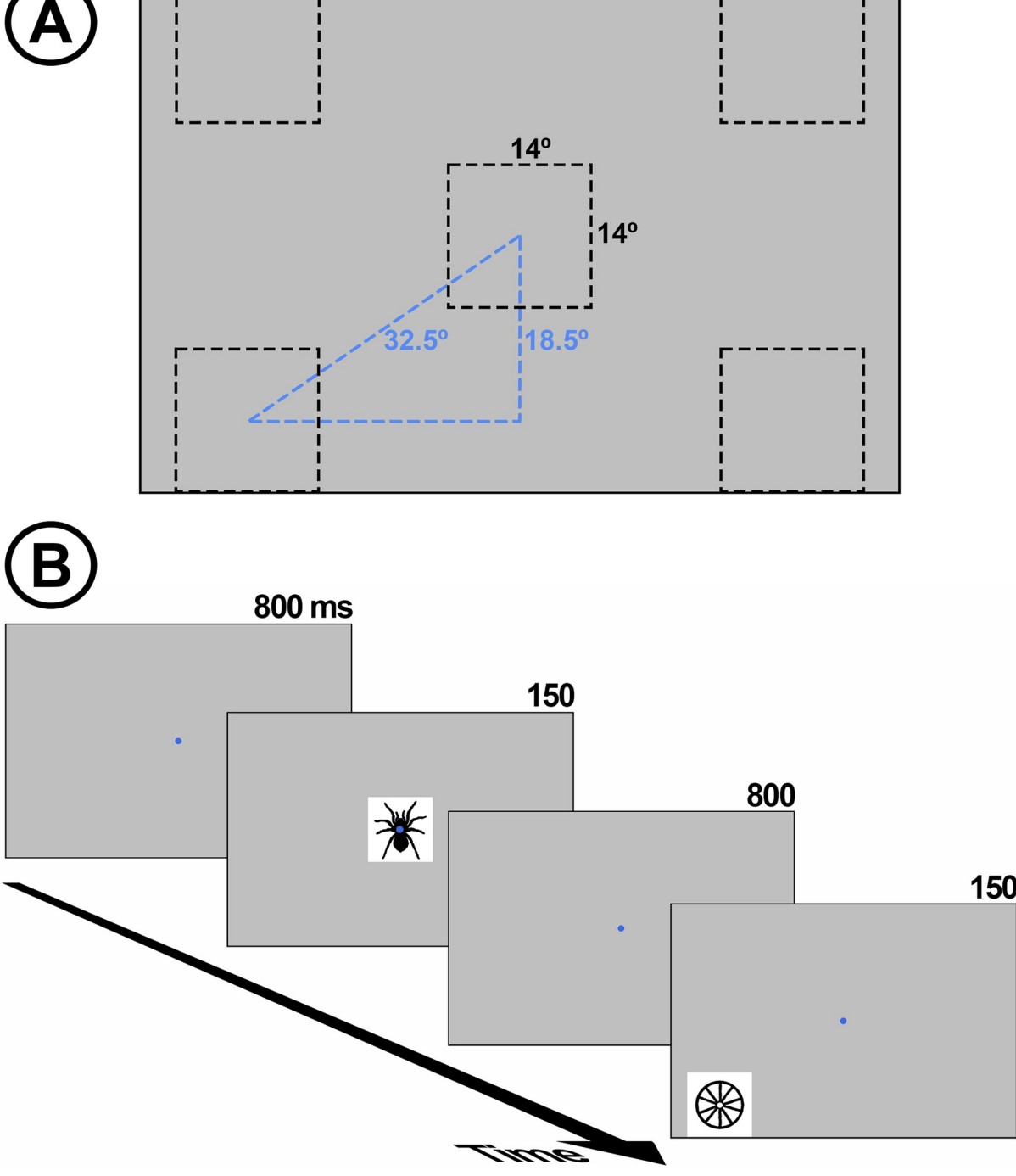

**Fig 1. Experimental design. A:** Size and possible locations of the stimulus in each trial. **B:** Schematic representation of one portion of the stimulus sequence. One of the exemplars of spider and wheel probes are depicted, each in one of the five possible spatial locations.

Two types of stimuli were presented to participants (Fig 1): 20 emotional silhouettes (spiders) and 20 neutral (wheels), all in black color over a white background. The size of stimuli (figure + ground) was 14° x 14° width. Spiders are among the top five most feared animals [37] and they cause the most prevalent phobia related to animals [38]. Indeed, spiders are

assessed as negatively valenced stimuli by relatively large samples in emotional picture data-bases (e.g., IAPS: [39]; EmoMadrid: [40]). In order to test whether spider *silhouettes* were also efficient as negatively valenced stimuli, and wheels as neutral, they were previously evaluated by an independent sample of 447 participants (397 women, mean age = 19.51, SD = 1.46) who rated their emotional valence through a 7-point Likert scale that ranged from "very negative" (1) to "very positive" (7). Spiders were rated as negative (mean = 1.704, standard error of means [SEM] = 0.038) and wheels as neutral (i.e., in the intermediate values of the scale: mean = 3.918, SEM = 0.030). Differences between both stimuli were strongly significant (F $(1,446)$ = 2557.289, p<0.001, $\eta^2_p$ = 0.852).

As mentioned in the introduction, stimuli presenting Gestalt characteristics such as closed contours or compact shape, as it is the case of silhouettes, are optimal to increase the response of contour-sensitive neurons present in V1 and V2 [34]. Moreover, the use of black silhouettes over white background inherently equalizes color and contrast, which may influence early visual ERPs (color: [41]; contrast: [42]), across experimental categories. Luminosity (i.e., figure surface against background) and spatial frequency of silhouettes, which may also influence ERP components of interest (luminosity: [43]; spatial frequency: [44]), were manipulated so they did not significantly differ between categories (spiders vs. wheels). Details on these two low-level characteristics and statistical contrasts, as well as the stimuli themselves, are provided in EmoMadrid (https://www.psicologiauam.es/CEACO/EmoMadrid/EMsiluetas.htm). In sum, the only visual parameter besides their emotional meaning clearly differing among spi-ders and wheels was their shape, in any case sharing certain key characteristics (e.g., wheel spokes may resemble spider legs and vice versa). More importantly, shape per se has been reported to firstly affect ERPs in latencies longer than those explored in this study [45–47].

## Procedures

**Experiment 1.** Each spider and wheel appeared 10 times in random order in one of the five locations depicted in Fig 1, one at fixation (FIX) and four peripheral (the center of each peripheral position was 32.5º from the center of the screen). Peripheral positions were upper-left visual field (UL), upper-right (UR), lower-left (LL), and lower-right (LR). This resulted in 200 trials per emotional category and location (20 exemplars x 10 presentations), and the total number of trials was 2000 (200 x 2 categories x 5 locations). Each stimulus, whatever its loca-tion, was displayed on the screen for 150 ms, and the inter-trial interval (ITI) was 850 ms. Par-ticipants were instructed to look at the fixation dot at the center of the screen all the time, which was marked with a blue circle (0.3˚ radius, RGB = 0, 0, 255) during the interstimulus intervals. The total duration of the whole stimulus sequence was ≈32 minutes, so it was divided into eight blocks to provide brief rest periods. In order to engage constant attention to stimulation, the inter-stimulus fixation dot randomly changed its color from blue to red (255, 0, 0) in 1 to 5 trials per block (0.5–2.5% trials per block), and participants were instructed to mentally count these changes and report the total number after each block (this sum was dif-ferent from block to block). None of the participants deviated more than one color change from the correct answer per block. As explained later, each red dot trial, and the next, were removed before analyses.

**Experiment 2.** Some methodological implementations were introduced in Experiment 2. First, further increasing the number of trials was considered a priority to boost SNR: each cate-gory (spiders and wheels) was presented 500 times (vs. 200 in Experiment 1). In addition, and also to increase this ratio, a jitter was added to the ITI to avoid a phase-locking of alpha EEG activity with the stimulus presentation rate. Thus, the ITI presented five different durations (750, 800, 850, 900, 850 ms), the average being 850 ms (as Experiment's 1 fixed ITI). Third, we

changed the color of the fixation dot. In Experiment 1, the contrast between the fixation dot (dark blue) and either the figure (black) or the background (white) was unbalanced between spiders and wheels when they were presented at fixation: the dot was surrounded by the figure -black- in 85% (i.e., less contrast) of the spider trials and 25% of the wheel trials. Thus, we presented the same stimuli in the replication but the fixation dot presented a grey color (RGB: 128, 128, 128) equidistant from white (255, 255, 255) and black (0, 0, 0) to discard any influence of this dot-stimulus contrast in the observed effects. And fourth, Experiment 2 presented stimuli only at fixation given that peripheral stimuli failed to elicit the N40 component and to show significant differences between spiders and wheels in P80 in Experiment 1, as later described and discussed in the Results and Discussion sections.

The stimuli were identical to those employed in Experiment 1, and maintained the same size and duration, but were presented only at fixation as indicated. As in Experiment 1, the presentation order was random and the total run ($\approx$ 16 minutes) was divided into four blocks. In each of them, the black fixation dot changed to red instead of grey in 1 to 5 trials per block (0.5–2.5% trials per block) and the task consisted again in "mentally counting" the number of changes to red and reporting the number of changes at the end of each block. The data from one of the discarded participants (see Participants section) were not included due to a deviation by more than one from the correct number of changes in one of the blocks (the rest of the participants did not exceed this deviation limit in any block).

## Recording and pre-processing (common to Experiments 1 and 2)

Electroencephalographic (EEG) activity was recorded using an electrode active cap (Biosemi®) with Ag-AgCl electrodes, in which the EEG signal is preamplified at the electrode. Sixty-four electrodes were placed at the scalp attending a homogeneous distribution (see Supplemental material) and the international 10–20 system. Following the BioSemi design, the voltage at each active electrode was recorded with respect to a common mode sense (CMS) active electrode and a passive electrode (DRL) replacing the ground electrode. All scalp electrodes were referenced offline to the nosetip. Electrooculographic (EOG) data were recorded supra- and infraorbitally (vertical EOG) as well as from the left versus right orbital rim (horizontal EOG) to detect blinking and ocular deviations from the fixation point. An online analog low-pass filter was set to 104Hz (5th order, CIC filter), with no high-pass filter. Recordings were continuously digitized at a sampling rate of 512 Hz. An offline digital Butterworth bandpass filter of 0.01 to 30 Hz (2nd order, zero-phase forward and reverse–twopass- filter) was applied to continuous (pre-epoched) data using the Fieldtrip software (http://fieldtrip.fcdonders.nl; [48]). Setting the high-pass filter at 0.1 Hz or less has been recommended to study early ERP components [49]. The continuous recording was divided into 300 ms epochs per trial, beginning 100 ms before the probe stimulus onset.

EEG epochs corresponding to trials in which the fixation dot changed its color (see the previous section) were eliminated, as well as those corresponding to the subsequent trial, to avoid the effect of this control, irrelevant (to our scopes) task. Blinking-derived artifacts were removed through an independent component analysis (ICA)-based strategy [50], as provided in Fieldtrip. After the ICA-based removal process, a second stage of inspection of the EEG data was conducted to automatically discard trials in which any EEG channel surpassed ±100 µV and/or its average global amplitude (i.e., maximum minus minimum amplitude) across trials ± 3.5 standard deviations. The minimum number of trials accepted for averaging was 150 trials per participant and condition (i.e., each category presented in each location). Data from six of the discarded participants in Experiment 1 (see Participants section) were eliminated since they did not meet this criterion, and the other two had to be discarded

because of data storage issues. In Experiment 1, this trial and participant rejection procedure led to the average admission of 179 ($SD = 7$), 180 (8), 181 (8), 182 (8), and 181 (7) trials at each of the five locations in the case of spiders, and of 181 ($SD = 8$), 181 (7), 180 (8), 181 (7), and 180 (9) in the case of wheels, the difference among stimulus categories being non-significant ($F(9,315) = 0.915$, $p = 0.512$, $\eta^2_p = 0.025$). For Experiment 2, the criterion was set at 400 trials minimum and three of the discarded participants did not meet it. In this experiment, the trial and participant rejection procedure led to the average admission of 451 ($SD = 16$) in the case of spiders and of 452 ($SD = 15$) in the case of wheels, the difference among stimulus categories being non-significant ($F(1,34) = 0.137$, $p = 0.714$, $\eta^2_p = 0.004$).

## Data analysis

**Experiment 1.**   First, recordings were baseline-corrected using the 100ms prestimulus interval. Next, we proceeded to identify and quantify the first visual component of ERPs. As illustrated in Fig 2 (where only the final 50ms portion of the baseline appears for graphical purposes; grand averages including the whole baseline are available in Supplemental material), an N40 component is visible in grand averages in response to FIX stimuli, being less evident in response to peripheral stimuli. To objectively confirm the existence or not of the N40 component, we determined whether amplitudes greater than typical baseline amplitude existed in its corresponding time window considering, at the same time, that N40 typically presents a very low SNR so too stringent criteria could mask this component. Thus, for each condition and within the 30–60 ms interval, the occurrence of N40 was confirmed when at least two neighbor channels (within the relevant scalp region, i.e., the posterior hemiscalp) presented at least two consecutive voltage points whose amplitude was beyond ±1.5 times the standard deviation of the corresponding baseline. This procedure revealed that N40 took place in the FIX conditions (both spiders and wheels), but not in any of the remaining eight conditions (four peripheral locations x two types of stimuli). To define N40 peak latency in response to FIX stimuli, recordings at parietal and occipital electrodes, bilaterally, were averaged together to provide a meta-average (Fig 2). The latency of the most negative value of the meta-average between 30 and 60 ms was defined as the N40 peak, which was 39 ms. Therefore, N40 amplitude to FIX stimuli was individually quantified as the average amplitude within the 36 to 42 ms window of interest (WOI).

The next component in time (P80) was also detected and its amplitude quantified in all conditions, since it was patent in all of them, including peripheral (Fig 2). Thus, P80 peak latency was defined by averaging together recordings at parietal and occipital electrodes, bilaterally in the case of FIX stimuli and contralaterally for peripheral stimuli, to obtain meta-averages (Fig 2). The latency of the most positive value of meta-averages between 60 and 110 ms was defined as the P80 peak: 80 ms for FIX stimuli (WOI to compute individual amplitude: 74–86 ms), 88 for LL (WOI: 82–94 ms), 86 for LR (WOI: 80–92 ms), 90 for UL (WOI: 84–96 ms), and 101 for UR (WOI: 95–107 ms): Table 1.

We also measured the differential N40-P80 amplitude, or peak to peak amplitude, a classical way of computing amplitudes (e.g., [51–53]) that has recently been revealed as useful to explore early visual ERP components [7]. Two advantages of this measure may be underlined. First, it is less susceptible to be affected by data processing settings such as the high-pass filter cut frequency, which significantly affects traditional (monophasic) amplitude measures in early components [49, 54] or the length of the baseline, another critical aspect in this regard. Second, it allows to quantify neural processes transversally affecting neighbor components by eliciting an increase of absolute amplitude in both of them. To this aim, the individual difference between P80 and N40 amplitudes (each computed as explained above) was calculated for FIX stimuli (Fig 2), given that peripheral stimuli did not elicit the N40 component.

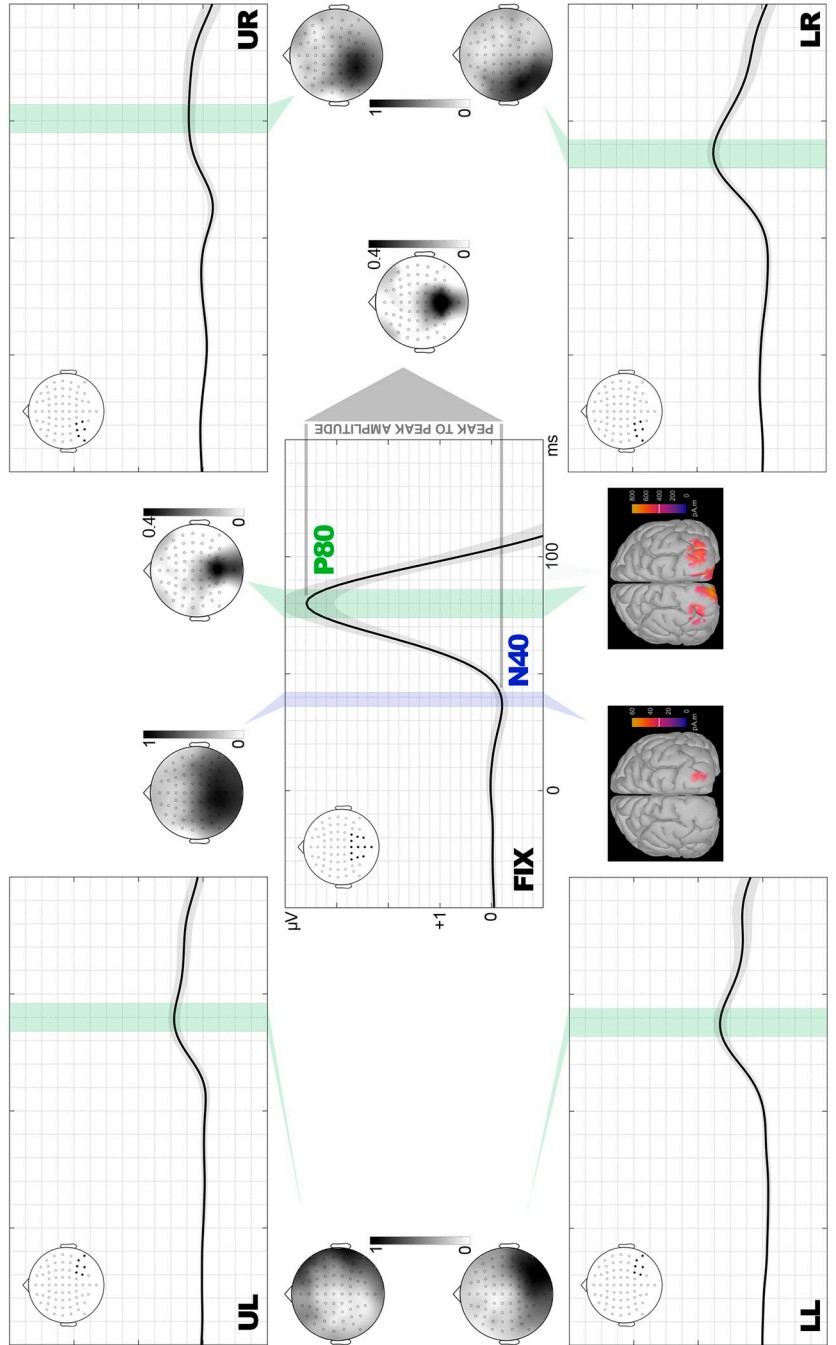

**Fig 2. Experiment 1: Windows of interest (WOI) and their outputs, sPCA and source estimation.** WOIs for N40 (blue bar), only found in FIX, and P80 (green bars), patent in all conditions, are represented over meta-averages computed from the electrode sites marked in black in each scalp map (FIX: fixation, LL: lower left, LR: lower right, UL: upper left, UR: upper right). Temporal and amplitude scales are the same for all locations and are defined in FIX. Shadows surrounding meta-average lines represent the standard error of means. Topographic maps of sPCA-derived relevant factor scores corresponding to each WOI are also depicted, as well as source estimations corresponding to WOIs of FIX conditions, which were those finally showing significant effects (see Supplemental material for source estimations in the rest of conditions).

**Table 1. Experiment 1: Main analytical outputs.**

|  |  | Peak | WOI | SFs (% var) | T(35) | p | Cohen d's | BF10 (Bayes) |
|---|---|---|---|---|---|---|---|---|
| N40 | FIX | 39 | 36–42 | 5 (92.7) | **-1.747** | **0.045** | **-0.291** | 1.342 |
| P80 | FIX | 80 | 74–86 | 5 (92.7) | **3.525** | **<0.001** | **0.588** | **53.470** |
|  | LL | 88 | 82–94 | 4 (87.5) | -0.589 | 0.720 | -0.098 | 0.121 |
|  | LR | 86 | 80–92 | 5 (88.8) | -1.582 | 0.094 | -0.264 | 0.075 |
|  | UL | 90 | 84–96 | 4 (86.6) | -2.134 | 0.980 | -0.356 | 0.062 |
|  | UR | 101 | 95–107 | 6 (88.9) | -0.743 | 0.769 | -0.124 | 0.110 |
| N40-P80 | FIX | - | - | 5 (93.5) | **4.547** | **<0.001** | **0.758** | **764.941** |

Peak latencies (in milliseconds), windows of interest (WOIs), number of spatial factors (SFs) extracted in sPCA, total variance they explain (% var), and outputs of both Student's frequentist and Bayesian t-tests on the spider>wheel difference corresponding to N40 and P80 amplitudes (FIX: fixation, LL: lower left, LR: lower right, UL: upper left, UR: upper right).

To avoid the multiple comparison problem that analyses are potentially affected by, the experimental effects on N40 and P80 were analyzed by submitting their amplitudes in the corresponding WOIs to a spatial principal component analysis (sPCA) on SPSS 26.0 [55]. This procedure reduces the electrode information (64 levels) into a small number of spatial factors (SFs) explaining, for the whole experimental sample, most of the variance due to the scalp location of recordings. Importantly, principal component analysis has been long defended as a preferable methodology to detect and quantify ERP components over traditional methodologies (e.g., [56–59]). In the space domain (sPCA), the main advantage of PCA over classical procedures based on visual inspection of topographies to define regions of interest is that it presents each ERP component separately and with its 'clean' shape, extracting and quantifying it free of the influences of adjacent or subjacent components. Indeed, several neural processes (and hence, several electrical signals) may concur at any given moment, and the recording at any scalp location at that moment is the electrical balance of these different neural processes. Such recording can stymie visual inspection. Spatial PCA (sPCA), in which variables are the electrodes and cases are participants x conditions, separates ERP components along space, each spatial factor ideally reflecting one of the concurrent neural processes occurring at any given moment or temporal interval. Additionally, sPCA provides a reliable division of the scalp into different recording regions. Basically, each region or spatial factor (SF) is formed with the scalp points where recordings tend to covary. As a result, the shape of the sPCA-configured regions is functionally based, and scarcely resembles the shape of the geometrically configured regions usually defined by traditional procedures. The spatial factor score, the sPCA-derived single parameter (per participant and condition) in which each SF is quantified, "summarizes" the behavior of the whole set of electrodes it involves (with different weights) and is linearly related to original amplitudes.

A separate sPCA was applied to each stimulus spatial location (FIX, LL, LR, UL, UR) given i) that N40 was only elicited by FIX stimuli and ii) that the latency of P80 varied across locations, as indicated. Components were selected based on the scree test and subsequently submitted to varimax rotation, which provides optimal performance in sPCA [60]. Factor scores corresponding to those SFs showing a parietal/occipital distribution (bilateral for FIX conditions or contralateral to stimulus location for peripheral conditions), the one relevant as regards early visual ERPs, were then submitted to statistical contrasts. A double contrast strategy was carried out using JASP software [61]. First, a one-tailed (given that our scope was detecting the earliest trace of sensory gain -i.e., greater activity in visual processing structures-towards emotional stimuli) frequentist repeated-measures Student's T-test was carried out

introducing Emotion of the probe (spiders, wheels) as factor. Effect sizes in these tests were computed using the Cohen's d formula. Second, Bayesian paired samples T-tests using the default prior (0.707), corresponding to medium effect sizes, were carried out on the same data to test the likelihood of data on H1 (spider > wheel) over H0 (spider = wheel) (BF10).

Finally, and to better characterize N40 and P80, their sources were estimated via the Minimum Norm (MN) method using the current density map algorithm as implemented in Brainstorm, v2021 [62]. To this aim, average amplitudes within the WOIs of each component showing significant effects in the previous (statistical contrast) step were submitted to this algorithm (depth weighting order and maximal amount: 0.5 and 10, respectively; noise covariance regularization: 0.1; SNR: 10), which was applied on a realistic cortex model defined through the openMEEG package [63, 64]. However, source estimations of the P80 component recorded in conditions showing insensitivity to the experimental treatment (i.e., peripheral conditions) are also available in Supplemental material.

### Experiment 2

After baseline (100 ms) correction, the first analytic task was identifying and quantifying the first visual component of ERPs. As may be appreciated in Fig 3 (where only the final 50 ms

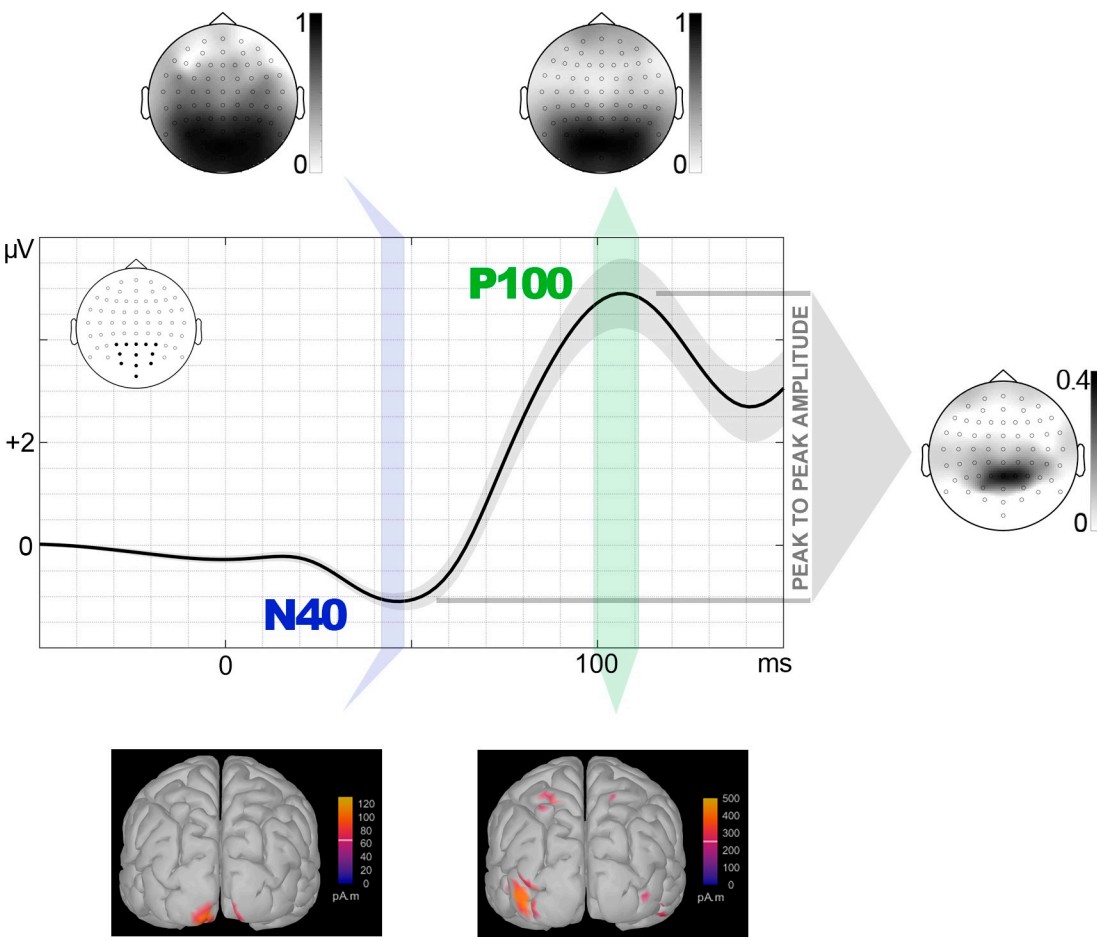

**Fig 3. Experiment 2: Windows of interest (WOI) and their outputs, sPCA and source estimation.** WOIs for N40 and P100 (green bars) are represented over meta-averages computed from the electrode sites marked in black in the scalp map. Shadows surrounding meta-average lines represent the standard error of means. Topographic maps of the sPCA-derived relevant factor scores corresponding to each WOI and source estimations corresponding to WOIs are also depicted.

portion of the baseline is represented for graphical purposes; grand averages showing the complete baseline are available in Supplemental material), an N40 component is clearly visible in grand averages, so the confirmation procedure carried out in Experiment 1 was not necessary this time. To define N40 peak latency, recordings at parietal and occipital electrodes, bilaterally, were averaged together to provide a meta-average (Fig 3). The latency of the most negative value of the meta-average between 30 and 60 ms was defined as the N40 peak, which was 45 ms. Therefore, N40 amplitude was individually quantified as the average amplitude within the 42 to 48 ms WOI. The next component in time (P100) was also quantified after defining its peak latency, which was defined from the same meta-average just mentioned (Fig 3). Thus, the latency of the most positive value within the meta-average between 70 and 130 ms was defined as the P100 peak, which was 105 ms. The WOI defined to quantify the average amplitude of this component was 99 to 111 ms. We also measured the differential N40-P100 amplitude, or peak to peak amplitude. To this aim, the individual difference between P100 and N40 amplitudes, each computed as explained above, was calculated (Fig 3).

The same sPCA-based quantification method explained in Experiment 1 was also followed here on the WOIs corresponding to N40 and P100. Also, the double contrast strategy -frequentist and Bayesian- described in the previous experiment was again performed, using the same parameters, on factor scores yielded by sPCA introducing Emotion of the probe (spiders, wheels) as factor. Finally, the sources of N40 and P100 were estimated via the Minimum Norm (MN) source localization algorithm following the same specifications as in Experiment 1.

## Results

### Experiment 1

**N40 (stimuli at fixation).** The sPCA was computed on N40 amplitudes to FIX stimuli only, as explained above. This analysis yielded five SFs explaining most of the variance of the 64 electrodes (88.11%). SF2 was the one showing bilateral occipital/parietal distribution which, for obvious reasons, is the one relevant in this case. Figs 2 and 4 show the topography of N40-SF2, and Fig 4 depicts meta-averaged recordings from representative electrodes of this SF, along with descriptive plots. Its corresponding factor scores were then submitted to repeated-measures T-tests, both frequentist and Bayesian, on factor Emotion (spiders, wheels). As shown in Table 1, the former yielded significant differences (t(35) = -1.747, p = 0.045, d = -0.291), spiders showing more negative N40 scores/amplitudes (Fig 4). However, Bayesian analyses found only anecdotal evidence in favor of H1 (greater N40 amplitude -more negative- for spiders than for wheels): $BF_{10}$ = 1.342.

The MN source estimation analysis on N40 elicited by FIX stimuli was carried out on the average amplitude within the N40 WOI, as indicated. As illustrated in Fig 2, this analysis yielded V1 as one of the sources (concretely, the caudal apex of the calcarine sulcus), but also other foci at prefrontal areas (Table 2). This disparity of sources may point to a low SNR and to possible spurious solutions rather than to a spread cortical activation, an issue that will be discussed later and that was addressed in Experiment 2.

### P80 (stimuli at fixation and at the periphery)

P80 was clearly elicited by all stimuli, whatever their location, so sPCAs were applied to all conditions. Table 1 shows the number of SFs extracted for each stimulus spatial location in the case of P80, and their total explained variance, which was over 86% in all cases. Factorial loadings corresponding to those SFs showing occipital/parietal distribution, bilateral in the case of FIX stimuli and contralateral in the case of peripheral stimuli, which were those relevant to our scopes (i.e., SF5 for FIX, SF4 for UL, SF2 for UR and LL, and SF3 for LR), are represented in Figs 2 and 4.

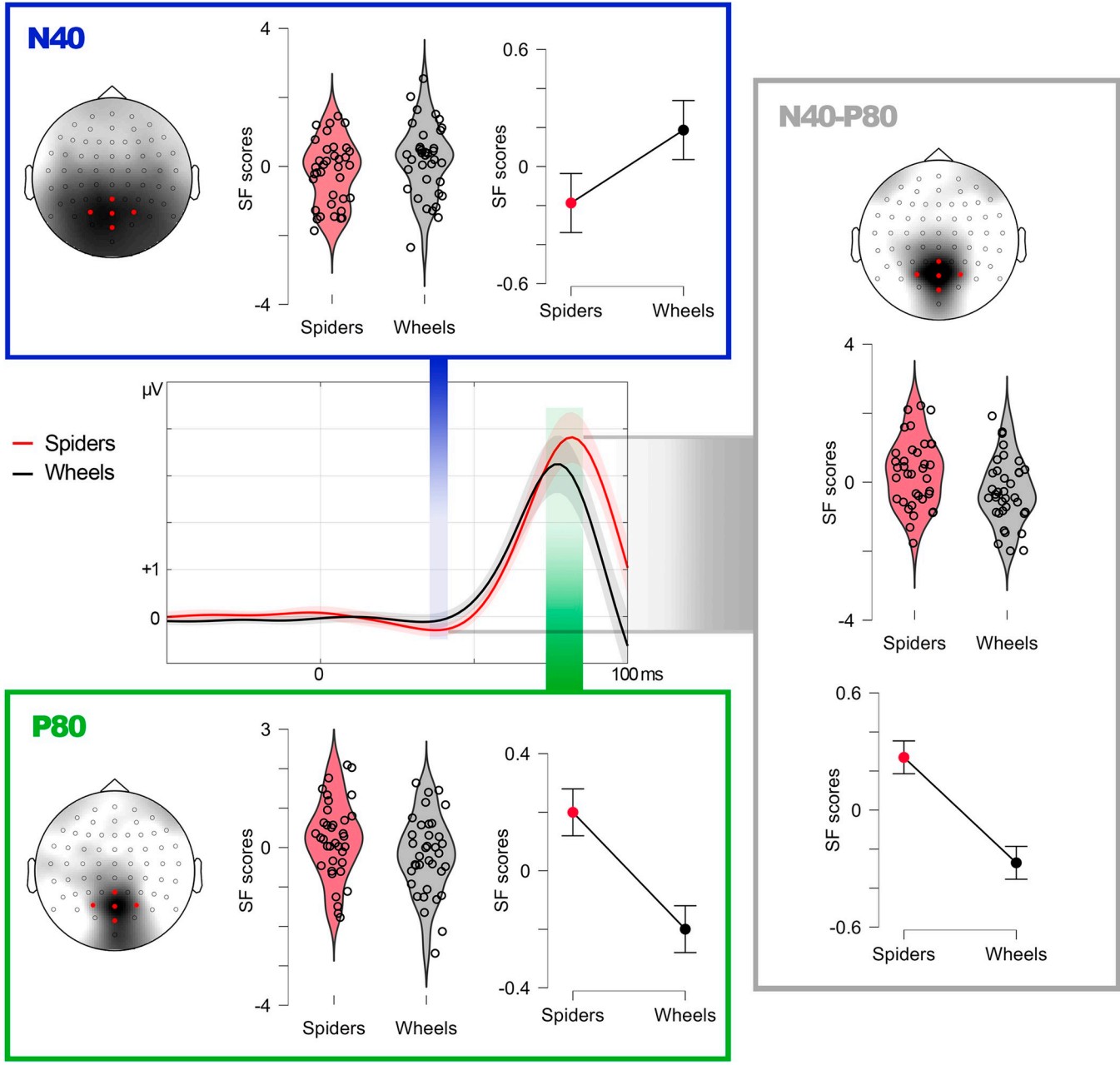

**Fig 4. Experiment 1: Descriptive data.** N40, P80, and N40-P80 peak-to-peak factor scores (linearly related to amplitudes) in response to spiders and wheels presented at fixation in relevant spatial factors. Violin plots show individual distribution and line graphs show means and standard error of means (error bars). For illustrative purposes, grand averages (center) are computed from five representative electrodes (marked in red) within the regions of maximal factorial load. Shadows surrounding grand average lines represent the standard error of means.

Factor scores derived from each SF were subsequently contrasted via repeated-measures T-tests on factor Emotion (spiders, wheels) and, since five contrasts were carried out (one per stimulus location), alpha was submitted to the Bonferroni adjustment procedure to avoid multiple comparison-derived type I errors. This adjustment set alpha at 0.01. Spiders elicited significantly greater P80 amplitudes than wheels when they were presented at fixation ($t(35) = 3.525$, $p<0.001$, $d = 0.588$): Fig 4. Instead, peripheral conditions did not yield significant spiders>wheels differences: Table 1. On the other hand, Bayesian analyses confirmed strong

**Table 2. Experiment 1: Source estimation.**

| N40 | | P80 | |
|---|---|---|---|
| **x, y, z** | **Anatomical label** | **x, y, z** | **Anatomical label** |
| 20, -100, 0 | Calcarine sulcus | -11, -105, -12 | Calcarine sulcus |
| 4, 73, 1 | SFG, frontal pole | 30, -101, 2 | Middle occipital gyrus |
| -55, 43, 7 | IFG, pars triangularis | | |

Main sources estimated through the Minimum Norm method for N40 and P80 in response to FIX stimuli (both spiders and wheels) and their peak MNI coordinates.

evidence in favor of H1 (spiders>wheels) in the case of FIX stimuli: $BF_{10}$ = 53.470, and null evidence (or even strong evidence in favor of H0) for peripheral conditions (Table 1). However, peripheral stimuli were actually perceived and discriminated, as revealed by later ERP components (these analyses and their results are described in Supplemental material for being out of the scope of this study).

Source estimation on P80 amplitude to FIX stimuli returned V1, bilaterally, as the main focus of activity (x = -11, y = -105, z = -12), along with bilateral foci in V2/LOC -lateral occipital cortex- (x = -30, y = -101, z = 2) with no other relevant foci in the rest of the cortex (V3, also present in this area, is an unlikely source since its main role is color processing): Table 2 and Fig 2. Supplemental material also includes source estimation of P80 to peripheral conditions, showing how main foci were located at visual cortices contralateral to stimulus location, more dorsally when presented in the lower visual field, and more ventrally for stimuli in the upper visual field.

## N40-P80 peak-to-peak amplitudes (stimuli at fixation)

Finally, N40-P80 differential amplitude was computed as the difference of the amplitudes of both components, each measured as indicated above, in response to FIX stimuli. These differences, calculated for each channel, condition (spider or wheel) and participant, were then submitted to a sPCA. The critical factor in this case was SF5 which, as illustrated in Figs 2 and 4, presented maximal loadings at midline parietal/occipital areas. The repeated-measures T-test contrasting its factor scores as a function of factor Emotion (spiders, wheels) yielded significantly greater N40-P80 peak to peak amplitude to spiders than to wheels, this result showing a large effect size (t(35) = 4.547, p<0.001, d = 0.758). The Bayesian repeated-measures T-test on these data found 'extreme' evidence in favor of H1 (spiders>wheels): BF10 = 764.941.

In order to test whether this significant sensitivity of N40-P80 peak-to-peak factor scores or amplitudes could be explained by either N40 or P80 alone (being in this case a redundant result), a repeated-measures ANCOVA was carried out using SPSS 26.0. In it, N40-P80 peak-to-peak amplitudes were introduced as the dependent variable, and N40 and P80 amplitudes, separately, as covariates. The covariates were both significantly related to the N40-P80 peak-to-peak amplitude [N40: F(1, 54.931) = 10.144, p = 0.002); P80(1, 62.322) = 133.905, p<0.001]. Indeed, the effect of Emotion of the probe on N40-P80 peak to peak amplitude is lost after controlling for both N40 and P80 amplitudes [F(1, 41.437) = 3.598, p = 0.065]. In other words, N40-P80 peak-to-peak effects depend on both N40 and P80 and reflects a neural process transversally affecting both deflections.

## Experiment 2

The sPCA computed on N40 amplitudes yielded five SFs explaining most of the variance of the 64 electrodes (89.87%). SF2 was the one showing a similar distribution to the relevant

N40-related spatial factor in Experiment 1. Neither frequentist (t(34) = -0.403, p = 0.345, d = -0.068) nor Bayesian analyses ($BF_{10}$ = 0.254) indicated significantly greater amplitudes for spiders than for wheels. As for P100, five SFs explained 93.40% of the variance, with SF3 presenting a similar distribution to that of the relevant P80 spatial factor in Experiment 1. Again, both frequentist (t(34) = -0.925, p = 0.819, d = -0.156) and Bayesian contrasts ($BF_{10}$ = 0.102) failed to find significant spiders>wheels differences. Figs 3 and 5 show the topography of N40-SF2 and P100-SF3.

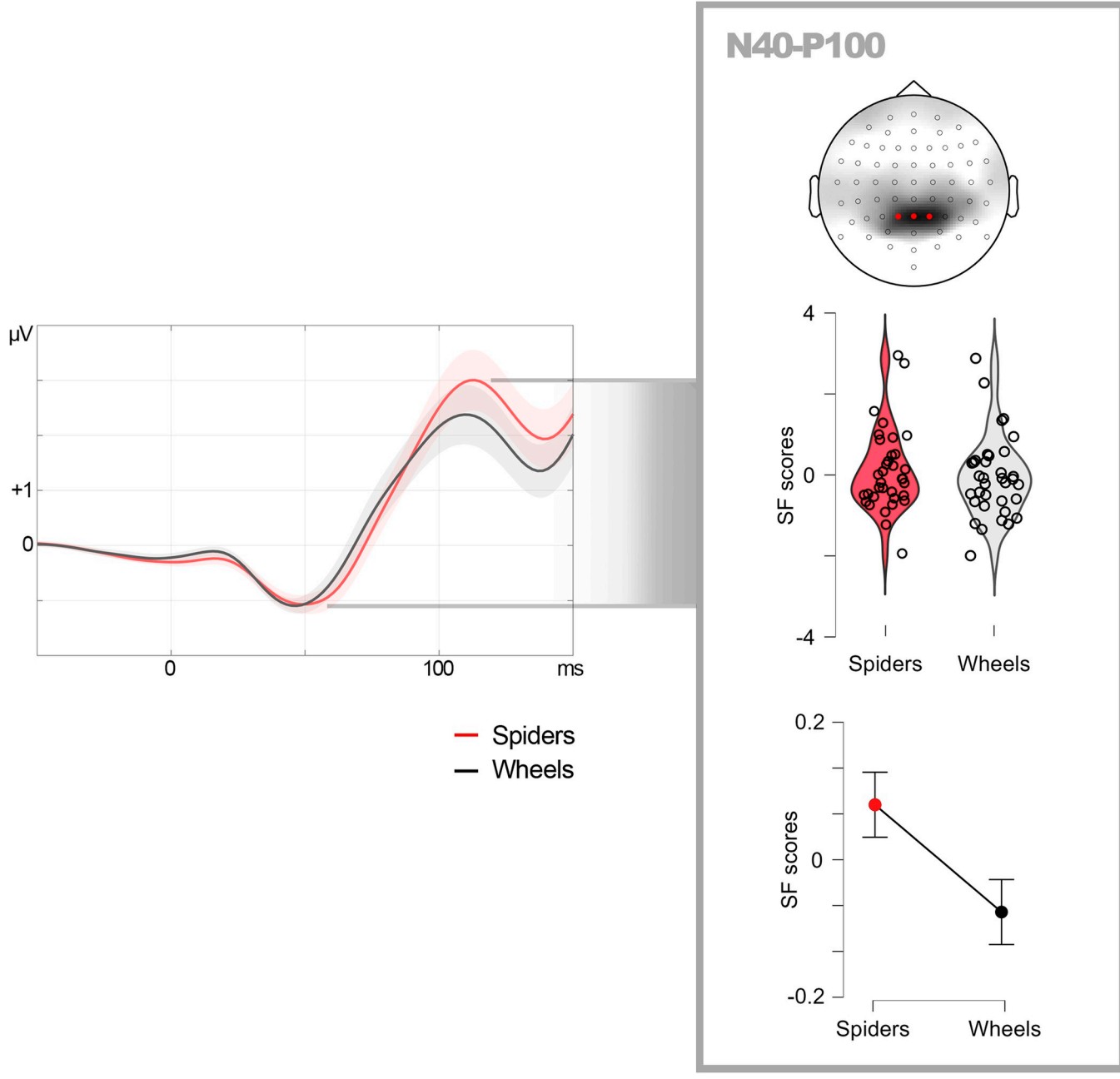

**Fig 5. Experiment 2: Descriptive data.** N40-P80 peak-to-peak factor scores (linearly related to amplitudes) in response to spiders and wheels in the relevant spatial factor. Violin plots show individual distribution and line graphs show means and standard error of means (error bars). For illustrative purposes, grand averages (center) are computed from three representative electrodes (marked in red) within the region of maximal factorial load. Shadows surrounding grand average lines represent the standard error of means.

**Table 3. Experiment 2: Main analytical outputs.**

|           | Peak | WOI    | SFs (% var) | T(34)  | p     | Cohen d's | BF10 (Bayes) |
|-----------|------|--------|-------------|--------|-------|-----------|--------------|
| **N40**   | 45   | 42–48  | 5 (89.9)    | -0.403 | 0.345 | -0.068    | 0.254        |
| **P100**  | 105  | 99–111 | 5 (93.4)    | -0.925 | 0.819 | 0.516     | 0.102        |
| **N40-P100** |   |        | 6 (96.1)    | **2.334** | **0.013** | **0.394** | **3.819**    |

Peak latencies (in milliseconds), windows of interest (WOIs), number of spatial factors (SFs) extracted in sPCA, total variance they explain (% var), and outputs of both Student's frequentist and Bayesian t-tests on the spider>wheel difference corresponding to N40 and P100 amplitudes.

As summarized in Table 3, N40-P100 peak-to-peak amplitudes did show significant effects, as in Experiment 1. Six sPCA components were extracted (explaining 96.111% of the variance) and, among them, SF6 showed a midline-parietal distribution similar to the relevant N40-P80 factor in Experiment 1. The frequentist contrast revealed significantly greater N40-P100 peak-to-peak amplitudes for spiders than for wheels (t(34) = 2.334, p = 0.013, d = 0.394), and also did the Bayesian test (BF$_{10}$ = 3.819). Fig 5 depicts meta-averaged recordings from representative electrodes within SF6, along with descriptive plots. As in Experiment 1, and to test whether this significant sensitivity of N40-P100 peak-to-peak factor scores or amplitudes could be explained by either N40 or P100 alone, a repeated-measures ANCOVA was carried out following the same procedure: N40-P100 peak-to-peak amplitudes were introduced as the dependent variable, and N40 and P100 amplitudes, separately, as covariates. The covariates were differently related to the N40-P100 peak-to-peak amplitude: while this relationship was significant in the case of N40 (F(1, 44.581) = 8.425, p = 0.006), it was not in the case of P100 (F(1, 64.022) = 1.386, p = 0.243). However, there was a significant effect of Emotion of the probe on N40-P100 peak to peak amplitude also after controlling for N40 and P100 individual amplitudes (F(1, 30.596) = 7.464, p = 0.010), suggesting additional mechanisms explaining this effect besides those reflected in these covariates.

MN source estimation analysis on N40 was carried out on the average amplitude within the N40 and P100 WOIs, as indicated (Table 4). As illustrated in Fig 3, this analysis yielded V1 as the net, main source (concretely, the caudal apex of the calcarine sulcus: x = -11, y = -105, z = -12). No other sources were observed in Experiment 2 for N40, probably due to the increased SNR provided by the methodological implementations previously mentioned. As for P100, the main source was located at V2/LOC (posterior part of the middle occipital gyrus: x = -45, y = -91, z = -3; see footnote 2). A second source for P100 was located in the superior parietal lobule (x = -26, y = -64, z = 68).

## Discussion

Previous studies place the earliest electrophysiological trace of emotional detection at around 80 ms from stimulus onset, concretely in the C1 component of ERPs [7, 17–19]. The two experiments confirm how quickly our visual system can detect certain emotional stimuli, but

**Table 4. Experiment 2: Source estimation.**

| N40 | | P100 | |
|-----|-----|------|-----|
| **x, y, z** | **Anatomical label** | **x, y, z** | **Anatomical label** |
| -11, -105, -12 | Calcarine sulcus | -45, -91, -3 | Middle occipital sulcus |
|  |  | -26, -64, 68 | Superior parietal lobule |

Main sources estimated through the Minimum Norm method for N40 and P100 and their peak MNI coordinates.

place this capability even earlier, starting at ≈40 ms from stimulus onset. This initial detection mechanism lasts up to ≈100 ms (P80 and P100 -which may be identified with traditional C1 and P1- in Experiments 1 and 2, respectively). The sources of this activity were located in visual cortices: V1 in the case of N40, both V1 and secondary cortices in the case of P80/C1, and mainly in secondary cortices in the case of P100/P1. This 40–100 ms window reflected in the N40-C1P1 peak-to-peak amplitude involves perceptual-attentional mechanisms implying interactions between those primary and secondary visual cortices which are transversal to the whole window rather than being circumscribed to a single ERP deflection. These results will be discussed in detail below, but it is important to underline at this point the novelty of these results as they are the first, to the best of our knowledge, to report the discrimination of emotional visual stimuli so early in time. This is understandable since pre-C1 activity, already little studied, has not been explored in response to emotional stimuli, either because studies analyzing this early visual ERP activity employed neutral stimuli or because experiments presenting emotional stimuli were not oriented or designed to record this activity.

As regards Experiment 1, the early capability of the brain to detect emotional stimuli was partially revealed by N40, which showed mixed evidence in frequentist and Bayesian contrasts and, robustly, by the N40-P80 peak-to-peak amplitude. This extremely fast activity reflecting the discrimination of emotional visual stimuli, and manifested in N40-P80, was statistically demonstrated to be due to both N40 and P80, and not to any of them separately. The visual cortex was found to be in the origin of both components. In the case of N40, solutions included V1. This first visual cortex stage receives the majority of inputs from the lateral geniculate nucleus of the thalamus [65]. Additionally, source estimation solutions unexpectedly included prefrontal areas, whose involvement seems improbable at this latency. These prefrontal foci likely reveal analytical noise and point to the desirability of increasing SNR in the second experiment. P80 sources were cleaner and also involved V1, along with V2 and/or lateral occipital cortex (LOC). Both V2 and LOC are progressively involved in object recognition, from contour and shape processing in V2 [66] to more global object identification in LOC [67].

These Experiment 1 effects were only observed for stimuli presented at fixation. However, spider > wheel differences in response to peripheral stimuli emerged in later, out of our scopes, ERP components, as shown in Supplemental material, demonstrating that these stimuli were actually perceived and evaluated. Two methodological factors may contribute to explaining the unexpected lack of sensitivity to peripheral stimuli of the earliest ERP components. First, SNR in early visual components is even lower for peripheral vision since it is underrepresented (compared to foveal vision), in terms of the number of neurons involved, both in the visual thalamus and in V1 (e.g.,[68]). Second, the task asked to direct attention towards fixation (i.e., color changes in the fixation dot). Considering that attention yields a biased competition at the perceptual level whereby limited processing resources prioritize attended spatial locations over unattended ones [69, 70], peripheral stimuli may have evoked diminished activity for this reason as well. These issues were beyond the scopes of the second experiment -focused on stimuli at fixation- to avoid an excessive number of trials, but deserve to be explored in the future.

Experiment 2 revealed similar early visual ERP components as those found in Experiment 1. Thus, both an N40 (presenting slightly higher latency: 45 ms) and a subsequent positive component, P100 (peaking at 105 ms) in this case, were evident at parietal and occipital regions. The variations in latency of both components with respect to Experiment 1 are probably due, along with the different sample of participants, to the implementations introduced in the experimental design. The most influential would be the variable ITI (instead of fixed), implemented to minimize alpha phase synchronization, which especially affects P1 latency

[71]. Importantly, Experiment 2 also confirmed the emotional effect on peak-to-peak amplitude involving both components, N40-P100 in this case. Thus, this amplitude was again significantly greater for spiders than for wheels. This replication reinforces the main finding in Experiment 1 and allows us to rule out that it was explained by possible confounding factors such as the contrast of the fixation dot over the background described in the Procedures section.

As in Experiment 1, this peak-to-peak amplitude effect was not explained by N40 or P100 separately, although the involvement of the former was stronger according to ANCOVAs. Moreover, neither N40 nor P100 showed significant effects when their single amplitude was analyzed. In this regard, Experiment 2 confirms the usefulness of analyzing peak-to-peak amplitudes in early visual ERPs, which appears to provide more complete and robust information (or less dependent on the experimental design) than single component analyses, at least when processes are transversal to two deflections rather than circumscribed to one of them, as seems to occur here. This classical way of measuring ERPs [51–53] has recently been revealed as useful for exploring early visual ERP components [7] and, as developed in the Data Analysis section, is less affected by signal processing procedures such as filtering or baseline definition.

Increasing SNR in Experiment 2 also allowed us to obtain cleaner source estimations, particularly in the case of N40. This time, the origin of this component was clearly located in V1, with no other appreciable sources. The origin of P100 was also the visual cortex, although the contribution of V1 was not as evident as in the case of P80 (Experiment 1), probably due to its longer latency ($\approx$25 ms). In line with previous studies, sources involved secondary areas [14], [72], concretely V2/LOC and the superior parietal lobule (SPL). The former source, involved in object recognition, was also observed and discussed in Experiment 1 with respect to P80, suggesting a -at least partial- functional link between P80 and P100. The SPL is a parietal area highly involved in attentional processes, both exogenous and endogenous, being a key node in the dorsal attention network [73]. This parietal area is consistently involved in attentional capture by emotional distractors (i.e., irrelevant to the task, as in this case;[74]). Moreover, this attentional capture by affective stimuli is typically reflected in P1, among other components [74]. Therefore, P100/P1 appears to reflect advanced stages of object identification (also observed in P80/C1, in Experiment 1), along with exogenous attention mechanisms.

The findings of both experiments, and particularly their timing, have several important implications at the theoretical level as regards emotional processing, particularly concerning the existing hypotheses on IESs. High-order structures such as the amygdala have been defended as a key IES capable of modulating the activity of the visual cortex, among other cerebral structures, at very short latencies (e.g., see reviews by [2]-[5]). However, the latency of the amygdala's enhanced response to emotional stimuli is not compatible with present data. Concretely, and according to intracranial EEG recordings, the earliest amygdala response to non-facial emotional visual stimuli is beyond 150 ms [8]. Moreover, even the amygdalar response to facial expressions, which is faster as indicated in the Introduction, does not modulate visual cortex activity until more than 250 ms later [6]. The alternatives to the amygdala hypothesis as an IES are currently under open debate. On the one hand, the "central position" postulates that the sensory cortex itself "is responsible for smart (fast and precise) initial evaluation of environmental threat" ([13]; p. 349). On the other hand, the "peripheral position" proposes that "beyond the central modulation of sensory experience, most sensory systems are tuned to conduct value-based appraisal of the environment before signals reach the cortex" ([75]; p. 917).

The results of our two experiments point to non-amygdalar candidates to be IESs and, whereas they are compatible with both the central and the peripheral alternative hypotheses, this key issue is worth being -at least briefly- discussed. Emerging evidence points to the

capability of earlier, first-order structures in the visual pathway such as the visual thalamus (see [9], for a review) or the superior colliculi ([12]; non-human data) to modulate their activity depending on the salience of the stimulus without the concourse of the visual cortex. Importantly, these structures modulate defensive behavior in response to visual stimuli in rodents (visual thalamus: [76]; superior colliculus: [77]). Concerning this, their abnormal activity has been proposed to be linked to affective problems such as anxiety or phobias in these same studies, pointing to their crucial role also at the clinical level. In our opinion, the key idea is that evaluation is a multistage process that requires all the steps (as each depends on the previous one), both rudimentary and precise, both fast and slow, to be "smart" and to allow for adaptive coping with emotional situations. In this chain of evaluative stages, the visual cortex, the amygdala, and other evaluative structures, would play a crucial role in different moments. However, as for the *initial* stage, which is the scope of this study, the peripheral hypothesis seems better positioned according to the scarce data available so far. In any case, further research is needed to advance this debate.

A remark on possible alternative interpretations and future directions should be made. While low-level visual parameters were controlled and homogenized between spiders and wheels, high-level differences apart from their emotional content exist, such as their semantic category (e.g., natural vs. artificial, or animal vs. object). Although this possibility cannot be discarded, we consider it very remote. Thus, while emotional stimuli are, by definition, relevant for the individual, the natural or animal condition of an item is orthogonal to relevance. For example, the relevance of a sparrow is minimal as compared to a snake for the majority of the population (as revealed by normative data in emotional picture databases cited above); in the non-animal/artificial category, an empty pot vs. a pistol pointing at us would be a parallel example. Critically, the evolutionary pressure on an extremely swift evaluation mechanism would be higher to detect emotion/relevance than to carry out a semantic categorization. Moreover, semantic processing of stimulation occurs later according to current data (e.g., first traces of animal vs. non-animal discrimination occur beyond 100 ms: [78, 79]. In any case, this extremely fast mechanism is worth being further explored by introducing additional stimulus categories, including emotionally positive items, and by manipulating the natural vs. artificial (or animal vs. non-animal) condition, to test the generalizability of current results. Relatedly, exploring individual differences is of great interest given that the activity of early evaluation structures is modulated by the individual experience thanks to the feedback they receive from other brain areas. For example, top-down modulation of visual cortices from evaluative structures higher in the hierarchy, such as the ventral prefrontal cortex [80], or of the visual thalamus from the visual cortex [81], plays a critical role in the activity of the initial evaluators. To conclude, this double-experiment study provides data on one of the less explored stages of affective processing and points to earlier-than-expected emotional evaluation processes.

## Author Contributions

**Conceptualization:** Luis Carretié.

**Data curation:** Luis Carretié.

**Formal analysis:** Luis Carretié, Dominique Kessel.

**Funding acquisition:** Luis Carretié.

**Investigation:** Uxía Fernández-Folgueiras, Guzmán Alba, Estrella Veiga-Zarza, Fátima Álvarez.

**Methodology:** Uxía Fernández-Folgueiras, Dominique Kessel, Fátima Álvarez.

**Resources:** Luis Carretié, Manuel Tapia.

**Software:** Luis Carretié, Uxía Fernández-Folgueiras, Dominique Kessel, Fátima Álvarez.

**Supervision:** Luis Carretié.

**Writing – original draft:** Luis Carretié.

**Writing – review & editing:** Uxía Fernández-Folgueiras, Dominique Kessel, Guzmán Alba, Estrella Veiga-Zarza, Manuel Tapia, Fátima Álvarez.

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
