## [Decision Letter · Decision Letter 0]

10 Apr 2024

PONE-D-24-05631An extremely fast neural mechanism to detect emotional visual stimuli: A two-experiment studyPLOS ONE

Dear Dr. Carretié,

Thank you for submitting your manuscript to PLOS ONE. After careful consideration, we feel that it has merit but does not fully meet PLOS ONE’s publication criteria as it currently stands. Therefore, we invite you to submit a revised version of the manuscript that addresses the points raised during the review process.

We look forward to receiving your revised manuscript.

Kind regards,

Yansong Li

Academic Editor

PLOS ONE

Journal Requirements:

2. Thank you for submitting the above manuscript to PLOS ONE. During our internal evaluation of the manuscript, we found significant text overlap between your submission and previous work in the [introduction, conclusion, etc.].

Please revise the manuscript to rephrase the duplicated text, cite your sources, and provide details as to how the current manuscript advances on previous work. Please note that further consideration is dependent on the submission of a manuscript that addresses these concerns about the overlap in text with published work.

[If the overlap is with the authors’ own works: Moreover, upon submission, authors must confirm that the manuscript, or any related manuscript, is not currently under consideration or accepted elsewhere. If related work has been submitted to PLOS ONE or elsewhere, authors must include a copy with the submitted article. Reviewers will be asked to comment on the overlap between related submissions (http://journals.plos.org/plosone/s/submission-guidelines#loc-related-manuscripts).]

We will carefully review your manuscript upon resubmission and further consideration of the manuscript is dependent on the text overlap being addressed in full. Please ensure that your revision is thorough as failure to address the concerns to our satisfaction may result in your submission not being considered further.

"This work was supported by Ministerio de Ciencia e Innovación [MICINN; grant number PID2021-124420NB-I00)."

4. hank you for stating the following in the Acknowledgments Section of your manuscript: 

"this work was supported by Ministerio de Ciencia e Innovación [MICINN; grant number PID2021-124420NB-I00)."

"This work was supported by Ministerio de Ciencia e Innovación [MICINN; grant number PID2021-124420NB-I00)."

Reviewers' comments:

Reviewer's Responses to Questions

**Comments to the Author**

1. Is the manuscript technically sound, and do the data support the conclusions?

Reviewer #1: Partly

Reviewer #2: Yes

2. Has the statistical analysis been performed appropriately and rigorously? 

Reviewer #1: Yes

Reviewer #2: Yes

3. Have the authors made all data underlying the findings in their manuscript fully available?

Reviewer #1: Yes

Reviewer #2: Yes

4. Is the manuscript presented in an intelligible fashion and written in standard English?

Reviewer #1: Yes

Reviewer #2: Yes

5. Review Comments to the Author

Reviewer #1: Presented experiment aimed to verify if the earliest waves of visual evoked potentials are modulated by the emotional load of visual stimuli. Responses to neutral pictures (wheels) were compared to responses to emotionally negative pictures (spiders). Thanks to large number of stimuli presentations, very early, weak wave (N40) was detected followed by more commonly described positive waves ~80-100 ms post-stimulus. Peak-to-peak amplitude (N40-P80/P100) is reported to be larger for emotionally loaded stimuli.

I have two major reservation, one about the organization of the manuscript and the other about the complicated data analysis.

A) In my opinion it is not necessary and even disadvantageous to divide the manuscript according to “the two experiments”. These are not really two experiments, but one experiment with constant methodology and two participant groups submitted to variants of a stimulation procedure. You can avoid all the redundancy by describing common methodology in one paragraph; listing consecutive results and writing one Discussion instead of three of them.

B) Concerns regarding analytical approach

1) I am not at all convinced that PCA approach was really justified for this particular experiment. Data reduction was indicated as a reason to use it, but data reduction could have been done in a classical way by choosing one or few electrodes of interest in the occipital area and averaging them, just like a time-window of interest was selected on the timescale. Actually, you do it for the figures (the waveforms called “meta-averaged recordings”).

PCA was applied to amplitude values extracted from all 64 electrodes, while the N40 is expected to be detected in a few of them. It is a waste of a method power – PCA was working on the variance not related to the question of this experiment.

If there is any other benefit from PCA (above data reduction) it should be clearly indicated.

2) The other point is the description of the analysis and reporting of the results. I admit that I am not a hard-core neuroinformatics specialist (like most of putative readers of this report), and I had a difficulty understanding how the data was fed into PCA. There was only one analyzed measure (i.e. amplitude); it is not obvious how were defined the variables fed to PCA – electrodes? how were defined the samples? participants? How were the emotional conditions taken to account? were they fed as independent samples in one analysis (i.e. N40_sub1_wheel; N40_sub1_spider; N40_sub2_wheel; N40_sub2_spider; …)? If so, I would be afraid that between-subject variance could be much higher than between-condition variance, again compromising the merit of PCA analysis.

Components were selected based on the scree plot – but what was the criterion/ cut off eigenvalue for principal components (PC) acceptance? Only one (out of five substantial PCs) was chosen for further analysis based on the topography i.e. high loading scores into occipital electrodes – what was the level of variance explained by these PCs?

Other comments

Introduction

page 4 line 20 “Nonfacial, consciously perceived emotional stimuli have not been explored in this respect.”

ERP difference in 50-100 ms window was shown for emotional words:

Rellecke, J., Palazova, M., Sommer, W., & Schacht, A. (2011). On the automaticity of emotion processing in words and faces: event-related brain potentials evidence from a superficial task. Brain and Cognition, 77(1), 23–32. https://doi.org/10.1016/j.bandc.2011.07.001

page 4 line 22

“Although less frequently, visual ERP components before C1 have also been reported “

It is worth noting that such waves are detected during standard clinical VEP recording with the use of flash stimuli which evoke series of waves starting as early as 30 ms.

Odom, J. V., Bach, M., Brigell, M., Holder, G. E., McCulloch, D. L., Mizota, A., Tormene, A. P., & International Society for Clinical Electrophysiology of Vision. (2016). ISCEV standard for clinical visual evoked potentials: (2016 update). Documenta Ophthalmologica. Advances in Ophthalmology, 133(1), 1–9. https://doi.org/10.1007/s10633-016-9553-y

Methods

What was the software used for stimuli presentation? Haw was it communicated with EEG recording system.

You declare to record 64 channels according to international 10-20 system, however there is no single one and only electrode pattern for 64 channels. Electrode map could be added in supplementary files.

N40 detection: “the occurrence of N40 was confirmed when at least two neighbor channels (within the relevant scalp region, i.e., the posterior hemiscalp) presented at least two consecutive voltage points whose amplitude was beyond ±1.5 times the standard deviation of the corresponding baseline.”

The +/- sign is confusing – in a search for a negative wave, you accepted also positive deviations from the baseline level?

p11, line 18 “... eliciting an increase of absolute amplitude in both of them, as is apparently the case here (Figure 2).”

No amplitude increases are evidenced in figure 2. There is only one trace with hardly visible N40.

In this respect – have you tried setting logarithmic Y scale to highlight low values? Also, while I appreciate that presentation of ERP baseline is a good practice, in case of such a short window as you analyze and draw, baseline could be reduced (to <=25 ms) for enhanced visibility of post stimulus waves. Longer sweeps can be available in supplementary materials.

In Figure 3 (and 5) we see a corridor around VEP lines – what it is? why the same variance measure is not plotted in figures 2 (and 4)?

Why different electrodes are chosen if Fig 2 and Fig 3 for VEP averaging (“meta-averaging”)? For both cases they were chosen form the same PCA map, they should be the same.

Page 19, line 12 “...one of her parents…” you should not disclose the sex of this participant, “...one of the parents…” would be better.

Figures are informative but fig. 2-5 are drawn with too thin lines and with too small elements. They are unreadable when scaled to print size (at 600 DPI). Figures need thicker lines, bigger head plots, bigger fonts. On the other hand, some descriptions are proportionally huge (e.g. UL, LL, etc) and panel outlines very thick (~1 mm thick). Only the wave names have reasonable sizes.

Reviewer #2: The paper titled "An extremely fast neural mechanism to detect emotional visual stimuli: A two-experiment study" investigates the neural processes underlying the rapid detection of emotional visual stimuli. The authors present a series of experiments using event-related potentials (ERPs) to explore the early neural responses to emotionally negative (spiders) and neutral (wheels) stimuli. The study addresses an important and relatively unexplored area in the field of emotion processing, focusing on the rapid neural mechanisms involved in the detection of emotional visual stimuli. The paper is well-structured, with clear explanations of the methodology and results.

However, there is room for improvement in terms of a more comprehensive discussion of limitations, broader theoretical implications, and potential individual differences:

1. The study's focus on a specific type of emotional stimulus (spiders) and a neutral control (wheels) may limit the generalizability of the findings to other types of emotional visual stimuli. In addition, the lack of a direct comparison with other emotional stimuli (e.g., positive or mixed emotions) leaves open questions about the specificity of the observed neural responses.

2. The study does not address potential individual differences in neural responses, which could be important for understanding variability in emotional processing.

3. The paper could have included a discussion on the potential practical applications of the findings, such as their relevance to clinical populations or their integration into computational models of emotion processing.

These questions can be discussed in more detail during the discussion session.

6. PLOS authors have the option to publish the peer review history of their article (what does this mean?). If published, this will include your full peer review and any attached files.

Reviewer #1: No

Reviewer #2: No

---

## [Author Response · Author response to Decision Letter 0]

25 Apr 2024

Please see the attached Response to Reviewers document (which include tables and figures not possible to include here)

---

## [Decision Letter · Decision Letter 1]

6 May 2024

An extremely fast neural mechanism to detect emotional visual stimuli: A two-experiment study

PONE-D-24-05631R1

Dear Dr. Carretié,

We’re pleased to inform you that your manuscript has been judged scientifically suitable for publication and will be formally accepted for publication once it meets all outstanding technical requirements.

Kind regards,

Yansong Li

Academic Editor

PLOS ONE

Additional Editor Comments (optional):

Reviewers' comments:

Review Comments to the Author

Reviewer #1: The revised manuscript is much improved and I have no further major comments.

I have noticed unclear sentence (transition of pages 21/22) :" This initial detection mechanism remains until ≈100 ms (P80 and P100 -which may be identified with traditional C1 and P1- in Experiments 1 and 2, respectively)" -- remains WHAT?

I would also suggest to add the traditional analysis results (that were included in the rebuttal as an Appandix) to supplementary materials.

---

## [Editor Report · Acceptance letter]

14 May 2024

PONE-D-24-05631R1 

PLOS ONE

Dear Dr. Carretié, 

I'm pleased to inform you that your manuscript has been deemed suitable for publication in PLOS ONE. Congratulations! Your manuscript is now being handed over to our production team.

Kind regards, 

on behalf of

Dr. Yansong Li 

Academic Editor

PLOS ONE